# The Association of Ovarian Teratoma and Anti-N-Methyl-D-Aspartate Receptor Encephalitis: An Updated Integrative Review

**DOI:** 10.3390/ijms222010911

**Published:** 2021-10-09

**Authors:** Cheng-Yang Wu, Jiann-Der Wu, Chien-Chin Chen

**Affiliations:** 1School of Medicine, Chung Shan Medical University, Taichung 402, Taiwan; fantasy7386@gmail.com; 2Department of Pathology, Ditmanson Medical Foundation Chia-Yi Christian Hospital, Chiayi 600, Taiwan; cych02505@gmail.com; 3Department of Cosmetic Science, Chia Nan University of Pharmacy and Science, Tainan 717, Taiwan

**Keywords:** anti-N-methyl-D-aspartate receptor encephalitis, autoantibody, encephalitis, germ cell tumor, ovarian teratoma, ovary, paraneoplastic neurological syndrome, teratoma

## Abstract

Ovarian teratomas are by far the most common ovarian germ cell tumor. Most teratomas are benign unless a somatic transformation occurs. The designation of teratoma refers to a neoplasm that differentiates toward somatic-type cell populations. Recent research shows a striking association between ovarian teratomas and anti-N-methyl-D-aspartate receptor (anti-NMDAR) encephalitis, a rare and understudied paraneoplastic neurological syndrome (PNS). Among teratomas, mature teratomas are thought to have a greater relevance with those neurological impairments. PNS is described as a neurologic deficit triggered by an underlying remote tumor, whereas anti-NMDAR encephalitis is characterized by a complex neuropsychiatric syndrome and the presence of autoantibodies in cerebral spinal fluid against the GluN1 subunit of the NMDAR. This review aims to summarize recent reports on the association between anti-NMDAR encephalitis and ovarian teratoma. In particular, the molecular pathway of pathogenesis and the updated mechanism and disease models would be discussed. We hope to provide an in-depth review of this issue and, therefore, to better understand its epidemiology, diagnostic approach, and treatment strategies.

## 1. Introduction

### 1.1. Ovarian Teratoma

Ovarian teratomas are the most common ovarian germ cell tumors (GCTs), and among all teratomas, the most frequently occurring ovarian GCTs are benign, cystic mature teratomas (MTs) [1,2]. Most teratomas are benign unless a malignant somatic transformation occurs. However, malignant transformation is scarce [3,4]. The designation of teratoma refers to a neoplasm that differentiates toward somatic-type cell populations, typically including cell populations that would naturally derive from ectoderm, endoderm, and mesoderm [2].

The current classifications of teratomas are divided into MTs, MTs with malignant transformation, immature teratomas (ITs), and monodermal highly specialized teratomas (e.g., struma ovarii) [2,4]. First, MTs accounted for 90% of all ovarian tumors in premenarchal girls and 60% of all ovarian neoplasms in women younger than 20 years old [5]. MTs are composed of mature differentiated elements, and all three germ layers are represented, thus showing highly differentiated tissue and highly morphological heterogeneity [1]. Some suggested that the presence of rare microscopic foci of the neuroepithelium (which is used in the diagnosis and grading of ITs) can be ignored due to the excellent outcome and, therefore, regarded as MTs [6]. However, according to a recent study, such tumors should always be classed as “ITs” if immature neuroepithelium is seen to avoid inappropriate classification and therapy due to vague cut-offs in different morphology [4].

MTs account for more than 95% of all ovarian teratomas [7] and are the most common ovarian germ cell tumors in women’s second and third decade of life [2]. The clinical presentation of MTs ranges from asymptomatic to chronic or acute pelvic pain, and rare complications such as cyst rupture and malignant transformation [8], denoting a degeneration of a somatic teratomatous element to a non-GCT malignant histologic type, equivalent to a somatic malignancy [3].

MTs with malignant transformation being the second classification of teratomas occur in 0.2 to 2 percent of mature cystic teratomas [2,9,10,11], comprising 2.9% of all malignant ovarian GCTs and 6% of GCTs [2,3,12]. Any of the components of an MT may undergo malignant transformation. However, squamous cell carcinoma arising from the ectoderm is the most common malignant transformation [2,13,14]. Others include well-differentiated neuroendocrine tumors, adenocarcinoma, sarcoma, and various rarer transformations of epithelial or soft tissue derivation. All require overgrowth of the organoid mixed nature of the MTs by a single element [2,4]. MTs with malignant transformation are aggressive tumors and typically resistant to conventional chemotherapeutic agents; thus, treatment must be tailored to the transformed histology [3].

The third classification of ovarian teratomas is ITs, known as malignant teratomas, embryonal teratomas, or teratoblastomas [2]. ITs comprise 35.6% of all malignant ovarian GCTs and less than 1% of ovarian teratomas [2]. ITs are commonly seen in the first two decades of life, yet the patients’ age ranges from younger than one year to 58 years [2,5,12]. Similarly, ITs can be composed of tissues from the three germ cell layers like MTs but arranged haphazardly and having varying amounts of immature tissue histologically [2]. ITs are the only ovarian GCTs to be histologically graded [2]. The grading is based on the proportion of immature neuroepithelial tissues that occupy the low-power field in any slides, ranging from “well-differentiated, grade 1” to “poorly-differentiated, grade 3” [4,5]. The grading system has its importance as being the indicator of the risk for extra-ovarian spread. Moreover, grade 1 ITs confined to the ovary do not require chemotherapy, whereas higher-grade ITs are needed [4]. The clinical manifestations of ITs are similar to other ovarian GCTs, primarily presenting adnexal or abdominal mass and pain. In addition, some patients may have mildly increased alpha-fetoprotein [2,5].

The last classification of teratomas is monodermal highly specialized teratomas, closely associated with MTs that consist of a predominant mature histologic cell type [2,15]. This rare and remarkable subset of teratomas may show a broad range of morphologies, such as struma ovarii, carcinoid neoplasms, sebaceous gland tumors, and neurogenic cysts [2,4]. The most common monodermal highly specialized teratoma is struma ovarii, representing approximately 3% of all ovarian teratomas, with thyroid tissues comprising more than 50% of the tumor mass [4,5], while the second most are ovarian carcinoid tumors [5]. Most patients present with abdominal mass, but the carcinoid syndrome resulted from the secretion of serotonin-like substances has been reported commonly in ovarian carcinoid tumors [5,16].

### 1.2. Paraneoplastic Neurological Syndrome (PNS) and Anti-N-Methyl-D-Aspartate Receptor (Anti-NMDAR) Encephalitis

PNS is defined as the pathologic involvement of the nervous system in the course of malignancy [17]. PNS is a heterogeneous group of disorders that may occur with any malignancy, although it is more commonly associated with small cell carcinoma, ovarian cancer, breast cancer, neuroendocrine tumors, thymoma, and lymphoma [18]. PNS may affect any central and peripheral nervous system level, either damaging one area or multiple areas [19,20]. The general diagnostic considerations of PNS are based upon the criteria defined by Graus et al. in 2004 [17,21]. These criteria use “classical syndromes of PNS,” “the occurrence of cancer,” and “onconeural antibodies” to make the diagnostic approaches [17,19,21]. The criteria then divide patients with suspected paraneoplastic syndromes into “definite” and “possible” categories [19,21].

The term “definite PNS” in contrast to “possible PNS” indicates a higher probability of paraneoplastic nature. Therefore, the former requires a more intense search for underlying tumors to reach early detection, while PNS often precede clinical manifestation of tumors [17]. By definition, the syndromes are caused by mechanisms other than metastases, tumor infiltration, metabolic and nutritional deficits, infections, coagulopathy, or side effects of cancer treatment [17,19,20].

Although the pathogenesis of PNS is not well understood, there is compelling evidence that PNS is caused by an immune response directed against neural antigens that are abnormally expressed by the tumor [19,20,22]. Immunologic factors are believed to play a pivotal role in the pathogenesis, especially autoimmune and T cell responses [19,20]. Onconeural antigens and their fragments that are aberrantly expressed by neoplastic cells are captured by tumor-associated dendritic cells and other antigen-presenting cells (APCs) through receptor-mediated endocytosis and phagocytosis [20]. Those APCs are then moved to regional lymph nodes to present the processed antigens to naïve T cells [20]. The binding of the T cell receptor to the MHC class II complex containing the antigen peptide in association with appropriate co-stimulatory signals induces the activation, differentiation, and proliferation of T cells. Then, activated CD4^+^ T helper cells induce the differentiation of B cells in the antibody-producing process [20,23]. Such an antibody has been deemed an autoantibody and directed against shared antigens ectopically expressed by the tumor or otherwise exclusively expressed by the nervous system [19].

Those neural-specific autoantibodies detected in PNS have been divided into two main categories according to the location of the target antigens, which are ”autoantibodies specific to intracellular antigens (e.g., cytoplasmic or nuclear)” and ”autoantibodies specific to plasma membrane antigens.” The two categories of antibodies can coexist [20]. The former is classical paraneoplastic or onconeural antibodies belonging to “well-characterized” paraneoplastic antibodies (Table 1). These antibodies are not regarded as pathogenic and serve as a surrogate marker of the PNS since their detection always indicates an underlying tumor. Moreover, in most of these disorders, the mechanism is believed to be mediated by cytotoxic T cells [19,20].

The other category comprises autoantibodies binding to the extracellular domain of proteins expressed by neuronal and glial cell surface or synaptic proteins (e.g., on muscle cells). In contrast to the first category, these antibodies have direct pathogenic potential on the target antigens [24]. These antibodies may occur with or without cancer, and the frequency varies according to different antibodies [19,24]. Some experts have also claimed that an underlying genetic predisposition might play a role in some of these disorders [19,25]. Examples of this category are listed in Table 2, including antibodies against the NMDAR. The pathogenic effect caused by anti-NMDAR antibodies is anti-NMDAR encephalitis, being an inflammatory condition of the brain.

As reported, anti-NMDAR encephalitis is defined as an immune-mediated disease characterized by a complex neuropsychiatric syndrome and the presence of cerebrospinal fluid (CSF) antibodies against the GluN1 subunit of the NMDAR [26,27]. Other manifestations are headaches, changes in mental status, seizures, language dysfunction, respiratory depression requiring ventilation, etc. [8,28]. Although this disease is rare, with an estimated 1.5 per million per year, anti-NMDAR encephalitis is the most common encephalitis associated with antibodies to cell surface antigens [18,27]. This disorder predominantly occurs in young women and children, although men and elders may also be affected [18,28]. The diagnostic criteria of anti-NMDAR encephalitis were developed in 2016 and primarily made by the detection of IgG antibodies to the GluN1 (also known as NR1) subunit of the NMDAR in serum or CSF, with the exclusion of recent history of herpes simplex virus encephalitis or other encephalitides, which might result in relapsing immune-mediated neurological symptoms (Table 3) [27,28]. In addition, anti-NMDAR encephalitis is associated with various tumors, such as MTs, mediastinal teratoma, small cell lung cancer (SCLC), and ovarian cystadenofibroma, while MTs are the most associated [8,28].

## 2. The Association of Anti-NMDAR Encephalitis and Ovarian Teratoma

The association between anti-NMDAR encephalitis and ovarian teratoma was first reported in 2007 [29]. While anti-NMDAR encephalitis comprised most autoimmune encephalitis, ovarian teratoma was associated primarily in female teens and adults and was reported as an underlying etiology in these encephalitis patients [30]. Furthermore, Dalmau et al. first facilitated the recognition of anti-NMDAR encephalitis and emphasized that autoimmunity can affect behavior, emotion, memory, and consciousness, particularly antibodies to the subunits of the NMDAR [29]. Since then, scientists and researchers have devoted themselves to better understanding the relevance between the PNS and teratomas. We have listed studies from 2007 to 2020 discussing the association and characteristics of female patients with ovarian teratomas and anti-NMDAR encephalitis and summarized some brief indexes and data according to the assessable cases (Table 4).

Regarding the clinical features of patients with metachronous anti-NMDAR encephalitis and ovarian teratoma, Bost et al. classified them into two groups by the coexistence of MTs or ITs. Although most reported encephalitis-related ovarian teratomas were MTs, ITs represented 11.8% of all ovarian teratomas in patients with anti-NMDAR encephalitis and were directly associated with a higher risk of death [33]. The patients with encephalitis and MTs were aged between 15 and 45 years (median age: 25), while the patients with encephalitis and ITs were aged between 12 and 38 years (median age: 22) [33]. It seemed to have a younger age distribution in the IT group. In addition, Dai et al. reported that the mean age of their patients with anti-NMDAR encephalitis was 23.14 ± 6.59 years [30]. Similarly, Zhang et al. observed that the mean age of the patients was 23.3 years, with the interval ranging from 14 to 36 years [34]. In summary, the overall mean age of all reported cases with metachronous anti-NMDAR encephalitis and ovarian teratoma was 23.97 years, ranging from 7 to 55 years (Table 4).

Histopathologically, most patients with anti-NMDAR encephalitis and ovarian teratomas had unilateral mature ovarian teratomas [29,30,47]. Regarding the laterality of ovarian teratomas, Acién et al. reported that ovarian teratomas occurred more frequently in the right ovary, but differences were not statistically significant [37]. In Table 4, there was no significant difference in tumor laterality since the total encephalitis-related ovarian teratomas were 72 right-sided, 61 left-sided, and 23 bilateral. Notably, the unilaterality comprised the majority [29,30,37,42]. Moreover, we found that the average size of encephalitis-related ovarian teratomas was 3.48 cm in our summary, while the smallest teratoma was only 1 cm. Iemura et al. observed that the tumor size of encephalitis-related ovarian teratomas was relatively smaller than control cases [48]. Interestingly, in patients with anti-NMDAR encephalitis, ovarian teratomas with immature foci were much larger (9 to 11 cm) than MTs [37].

The presence of nervous tissues in ovarian teratomas may play a role in pathogenesis. Chefdeville et al. reported that nervous tissue was present in 96% of anti-NMDAR encephalitis-associated teratomas, while only 38% of control MTs had nervous tissue (*p* < 0.001) [49]. In the same study, when neural elements were shown, there was a significant difference in the inflammatory infiltrates of B cells, T cells, and mature dendritic cells between anti-NMDAR encephalitis-associated teratomas and sporadic MTs [49]. The aggregation of lymphocytes within or around the neuroglial tissues may better understand the pathogenesis of ovarian teratoma-associated anti-NMDAR encephalitis [48,49].

In Table 4, we found that 64.7% of cases in ovarian teratoma with anti-NMDAR encephalitis would have prodromal symptoms and PNS. The prodromal symptoms are defined as signs that occurred before the first neurological symptoms, including headache, infection sign (fever, upper respiratory symptoms), and gastrointestinal symptoms (vomiting, diarrhea) [31,33]. The PNS exhibited diversity due to the broad impact at any central and peripheral nervous system level, causing manifestations such as mental and behavioral disorders, memory impairment, seizures, decreased consciousness, involuntary movement, speech disorder, autonomic dysfunction, and central hypoventilation or ventilator-assisted respiration. We briefly concluded some of most seen symptoms and their frequencies, which were behavior, personality disorders or psychosis; seizures; movement disorder; decreased consciousness; autonomic dysfunction; speech disorder; memory deficit; with a frequency of 89.8%, 82.4%, 79.6%, 77.1%, 70.5%, 60.4%, and 60.4%, respectively.

A relapse is defined as an exacerbation of previous symptoms or the onset of new symptoms after at least two months of improvement or stabilization [32,36]. Moreover, worsening symptoms were described as the modified Rankin Scale (mRS) increase ≥1 [30]. The mRS (Appendix A) is a widely used scale to evaluate the degree of disability or dependence in the daily activities of people who have suffered from stroke or other causes of neurological disability [50]. It is an established scale used in previous studies on anti-NMDAR encephalitis for an outcome, relapse, and response measurement [30,32,34,36]. We summarized the relapse rate as 7.1% in Table 4, which showed a lower frequency in previously reported studies (20–24%) [51]. This was probably caused by the better recognition of the disease, earlier treatment, and increasing use of the intervention. Notably, Titulaer et al. reported that patients who received second-line immunotherapy during the initial episode of encephalitis had fewer relapses [32]. 

The time at relapse also varied from studies [33,51]. Most patients with anti-NMDAR encephalitis experienced the first relapse within 24 months, but a relapse six years after onset was also reported. Other reports also suggested that anti-NMDAR encephalitis relapse could occur years after the initial episode [36]. In the study conducted by Zhang et al., they found that removing teratoma seemed critical to achieving final recovery, reducing the risk of relapse, and improving the long-term prognosis [34]. Interestingly, Yaguchi et al. observed a case that anti-NMDAR encephalitis occurred every time the teratoma relapsed, suggesting the association between encephalitis and teratoma [35].

Incidence is one of the main focuses we put our emphasis on. Generally speaking, women account for about 80% of patients with anti-NMDAR encephalitis, and many are accompanied by ovarian teratoma [29,35]. Investigating the incidence of ovarian teratoma among female patients with anti-NMDAR encephalitis, Dalmau et al. first reported that 11 of 12 female patients had ovarian teratomas [29], and Florance et al. observed that the frequency of ovarian teratomas was 56% in women >18 years old, 31% in girls ≤18 years old (*p* = 0.05), and 9% in girls ≤14 years old (*p* = 0.008) [31]. In a multi-institutional study, the incidence of ovarian teratoma was 44.2% among women with anti-NMDAR encephalitis [32]. The overall incidence, which we concluded in Table 4, was 37.4%. However, the majority of the incidence rates were calculated from “ovarian teratoma among anti-NMDAR encephalitis.” Little data was showing the incidence rate of “anti-NMDAR encephalitis among ovarian teratoma.” One single-institutional observational study found that female patients with anti-NMDAR encephalitis concomitant ovarian teratomas accounted for only 1.17% of all ovarian teratomas patients [35]. This is by far the only report showing the incidence of anti-NMDAR encephalitis in patients with ovarian teratoma. However, there might be an underestimation of anti-NMDAR encephalitis in patients with ovarian teratoma since diagnosing anti-NMDAR encephalitis needs experience and tests.

## 3. The Hypothetical Mechanisms and Models of the Pathogenesis

### 3.1. The Triggers and Peculiar Cell Composition in the Ovarian Teratomas with Anti-NMDAR Encephalitis

It is thought that anti-NMDAR encephalitis is a neuroinflammatory disease mainly mediated by autoantibodies against the GluN1 subunit of NMDA receptors [26,27]. However, the headstream triggers and disease association with ovarian teratoma were those we wanted to clarify. Two confirmed triggers of anti-NMDAR encephalitis were (1) tumors, which were primarily ovarian teratomas (with penetrance reported to be 94%) [8,28,52], and (2) postviral events, such as herpes simplex encephalitis [53,54]. This section specifically discussed the linkage between anti-NMDAR encephalitis and ovarian teratoma for a comprehensive understanding and increasing recognition. 

Zaborowski et al. hypothesized that the NMDARs might be expressed on the surface of ovarian teratoma cells [17]. Immune reaction against the NMDAR on tumor cells resulted in the production of anti-NMDAR autoantibodies [17]. Since ovarian teratoma cells were derived from human embryonic stem cells, the tumor had a haphazard arrangement of tissues recapitulating or resembling various somatic derivatives [55]. Thus, it was believed that components of teratomas might contain neuroglial cells and neuro-elements like NMDARs (Figure 1A). Chefdeville et al. reported that anti-NMDAR encephalitis-related ovarian teratomas had a greater prevalence of nervous tissue presentation than control sporadic ovarian teratomas, with the percentage of 96% vs. 38% and also shown significant difference (*p* < 0.001) [49]. Neuroglial tissues might be involved in triggering or sustaining the anti-tumor response associated with autoimmune encephalitis [49].

Moreover, Nolan et al. hypothesized that the generation of autoantibodies in anti-NMDAR encephalitis-related ovarian teratomas could be evidenced by investigating the neuroglial populations and the composition and topography of the involving immune cells (Appendix A) [56]. They observed that the density of mature neurons was reduced in anti-NMDAR encephalitis-related ovarian teratomas compared to control teratomas. At the same time, the co-localized neuroglial tissue and lymphoid follicle formation were more prevalent in anti-NMDAR encephalitis-related ovarian teratomas than controls. One potential explanation was that this represented the end-stage result of sustained autoimmune injury to the neurons. Consequently, a spectrum of damage ranging from varying degrees of degenerative changes to cell loss could be expected in anti-NMDAR encephalitis-related ovarian teratomas [56]. In addition, Jiang et al. observed that anti-NMDAR encephalitis-related ovarian teratomas showed more dysmorphic neurons with irregular cell shape and giant nuclei than those without encephalitis [52], suggesting that those dysplastic neurons being a potential source of auto-antigens could trigger anti-NMDAR encephalitis.

In conclusion, briefly, we could see ovarian teratomas with anti-NMDAR encephalitis having more frequent neuroglial components and, notably, more inflammatory infiltrates with over-representation of B cells, plasma cells, and dendritic cells, conforming tertiary lymphoid structures (TLS) (Figure 1A,B) [27,49,56]. In addition, evidence existed that tumor-infiltrating B cells could synthesize NMDAR antibodies in vitro, supporting the theory even more [57].

### 3.2. The Microenvironment Involving in Ovarian Teratomas with Anti-NMDAR Encephalitis

The schematic representation in Figure 1B shows a tertiary lymphoid structure located within an ovarian teratoma comprising a CD4^+^ T cell zone and CD20^+^ B cell zone with a germinal center, plasma cells, central memory cells, and antibodies against NMDARs and mature dendritic cells [58]. Remarkably, less frequent CD8^+^ T cells (T_C_) were reported on brain biopsy or autopsy in patients with ovarian teratoma and anti-NMDAR encephalitis [27,59]; thus, we inferred the less production of T_C_ at TLS and circulation in the bloodstream. Nonetheless, CD20^+^ B cells were more frequent in the teratoma-related anti-NMDAR encephalitis than in the controls (*p* = 0.001), suggesting a presiding humoral immune environment in those teratomas [52]. In addition, as shown in human autopsies, complement-mediated mechanisms were unrelated to the pathogenesis of teratoma-associated anti-NMDAR encephalitis [27,59,60].

The neural antigens (which were illustrated as antigens of NMDAR in Figure 1B) and their fragments were captured by APCs such as B cells and tumor-associated dendritic cells. Those APCs subsequently presented the processed antigens to CD4^+^ T cells via MHC class II tumor-derived peptide antigen, which induced T cells activation, differentiation, and proliferation. Activated CD4^+^ T helper cells (T_H_) induced the differentiation of B cells in the antibody-producing process, while the central memory T and B cells generated in TLS circulated systemically in the body (Figure 1B).

### 3.3. The Association with Teratoma-Related Anti-NMDAR Encephalitis and Blood–Brain Barrier Integrity

Immunological and pathological shreds of evidence have proven that NMDAR autoantibodies were synthesized systemically and within the central nervous system (CNS) by antibody-producing cells that can cross the blood–brain barrier (BBB) [27,61,62,63,64,65]. The central memory cells, plasma cells, and autoantibodies (specific IgG antibodies) circulated through the blood and lymphatic systems, then reached the CNS and eventually crossed the BBB. Nevertheless, the mechanism by which the autoantibodies cross the BBB is not yet fully understood [56]. One potential explanation was that the immune and inflammatory response in patients with ovarian teratoma and anti-NMDAR encephalitis might damage the integrity of capillaries and lead to the dysfunction of BBB [65]. Notably, Jiang et al. showed increasing levels of TNF-α, IL-10, and GM-CSF in patients with teratoma-related anti-NMDAR encephalitis compared to the control group, indicating the involvement of the inflammation process in the pathogenesis [52]. Moreover, the rising levels of GM-CSF not only up-regulated the production of pro-inflammatory cytokines but also formed positive feedback to activate microglial cells in the CNS, producing highly neurotoxic substances that might damage the coherence of BBB [65,66]. TNF-α had a moderating impact on BBB permeability via the internalization of tight junction proteins on endothelial cells [67], and GM-CSF was called for the recruitment of peripheral myeloid cells that contributed to blood–brain and blood–spinal cord barriers disruption [68], which eventually facilitated the transfer of serum antibodies through BBB to CSF [52].

Moreover, Table 4 demonstrates that around 70% of the patients had autonomic dysfunction, which may develop hypertension and sympathetic excitation that could increase the permeability of BBB. Some experts proposed that virus infection of CNS, especially herpes simplex virus encephalitis, may also trigger BBB dysfunction in anti-NMDAR encephalitis [69,70]. The theory correlated to our finding in Table 4 that almost two-thirds of patients with ovarian teratomas and anti-NMDAR encephalitis had experienced prodromal symptoms, which seemed like non-specific viral-like illnesses.

### 3.4. The Molecular Basis for Structures and Physiological Features of NMDAR

After the antibody-producing cells and autoantibodies entered the CNS, these antibodies were regarded as pathogenic in the brain as suggested by experiments using cultured neurons, exposing patients’ antibodies [27,71]. The IgG autoantibodies were then targeted by the same antigens of NMDAR, in which the receptors were widely distributed in the synaptic areas throughout the brain and spinal cord [17,72,73].

The NMDARs have continued to intrigue scientists over the years due to their crucial roles in CNS function. These ionotropic glutamate receptors were essential mediators of brain plasticity and can convert specific patterns of neuronal activity into long-term changes in synapse structure and function that are thought to underlie higher cognitive functions [72]. On the molecular basis, NMDARs were permeable to the physiologically relevant Ca^2+^, Na^+^, and K^+^ ions. The release of glutamate activated NMDARs at central synapses, which mediated an inward current (mainly the movement of Ca^2+^) and thereby depolarized the postsynaptic neurons, initiating excitatory postsynaptic currents (EPSCs) [74]. These currents depended on the membrane potential and frequency of synaptic release, rendering the NMDARs coincidence detectors that respond uniquely to the simultaneous pre-synaptic release of glutamate and postsynaptic depolarization with a current that allowed a substantial influx of external Ca^2+^ into the neuron’s dendritic spine [75]. The increase in intracellular Ca^2+^ served as a signal that led to multiple changes in the postsynaptic neurons, including changes that ultimately produced either short-term or long-term changes in synaptic plasticity (also known as synaptic strength) [76]. The varieties of synaptic plasticity depended on the frequency and duration of synaptic NMDAR activation, thereby providing the brain with a technique for encoding information [74,77]. Thus, the dysfunctions of NMDAR would involve various psychiatric and neurological disorders, as we reviewed in this article.

These NMDARs were heterotetramers composed of two obligatory GluN1 subunits, associated with two regulatory subunits of GluN2-type and GluN3-type, expressed in several divergent isoforms (GluN2A-D and GluN3A-B) [73,74]. Each semi-autonomous subunit was comprised of 4 domains shown in Figure 1D, which were the extracellular amino-terminal domain (ATD), the agonist binding domain (ABD), the transmembrane domain (TMD), which formed by three transmembrane helices (M1, M3, and M4) and a membrane reentrant loop (M2), and the intracellular carboxyl-terminal domain (CTD) [74]. All subunits shared a similar architecture which together formed a central ion channel pore. To be more precise, the channel pore was formed by TMD, with the reentrant loop lining the intracellular portion of the pore, whereas the transmembrane helices formed the extracellular region (Figure 1D). All of them were involved in the process of pore opening and accountable for the channel gating in NMDARs [54,74].

The distribution of different types of NMDARs also varied in different brain regions [73]. The GluN1 subunit, which bound glycine and D-serine, was obligatory in all functional NMDARs and widely expressed in virtually all central neurons [74]. As for the regulatory subunits, GluN2A and GluN2B, binding glutamate and by far the most abundant types in the mammalian brain, had their respective primary occupied regions [73]. The GluN2A-containing NMDARs were highly expressed in the adult hippocampus and cerebral cortex. Conversely, in other brain areas such as the striatum, GluN2B types were predominant [56,72,73]. Notably, the convincing locations for the cause of teratomas-related- encephalitis were at the hippocampus and prefrontal cortex of the brain (Figure 1) [56], which overlapped with the prevalent GluN2A-territory. From a more subtle point of view, GluN2A-NMDARs were more actively anchored in synapses in which their diffusion was relatively restrained and enriched in the postsynaptic density (PSD) compared to extra-synaptic sites [72,73,78]. In contrast, GluN2B-NMDARs were dispensed more extra-synaptic and explored larger dendritic areas [78].

### 3.5. The Role of NMDARs in Ovarian Teratomas with Anti-NMDAR Encephalitis

In anti-NMDAR encephalitis, the epitope to which the autoantibody bound was located at the amino-terminal domain (ATD) of the GluN1 subunit (Figure 1D) [61]. The IgG autoantibodies targeted surface GluN1-NMDARs and caused a selective and reversible decrease in GluN1-NMDAR surface density and synaptic localization [26,27]. Using real-time nano-particle tracking and imaging to elucidate the cellular dynamics involving NMDAR dysfunction, the incubation of patients’ autoantibodies significantly increased the mobile fraction of GluN2A-NMDAR and, in opposite manner, significantly decreased the mobile fraction of GluN2B-NMDAR [78]. It indicated synaptic GluN2A-NMDARs were mostly removed from synapses under the effects of autoantibodies, whereas extra-synaptic GluN2B-NMDARs were mainly cross-linked [78]. In addition, autoantibodies disrupted the surface interaction between NMDAR and Ephrin-B2 receptor (EphB2R), a synaptic protein which activation was shown to influence NMDAR trafficking and synaptic targeting [78,79]. This interference in synergy led to a lateral dispersal of synaptic EphB2Rs and NMDARs [78]. Furthermore, Paoletti et al. observed a specific dispersion of GluN2A subunits from synaptic sites [72], which also correlated with the above findings.

Recently, Jiang et al. reported that 90% of encephalitis-related teratomas were immune-reactive for antibodies directed against GluN2A and GluN2B epitopes [52], indicating that specific epitopes for autoantibodies were not restricted to only the GluN1 subunit. We considered the finding a more rational perspective since the particular areas of the affected brain were equitant to the specific distribution of GluN2 subunits, whereas the GluN1s were scattered evenly in nearly all CNS regions. Interestingly, the dysmorphic neurons in the encephalitis-related ovarian teratomas had strong immunoreactivity for NMDAR subunits GluN1, GluN2A, and GluN2B, and subsequent immunofluorescence showed consistent colocalization of NMDAR subunits with IgG [52]. This consolidated the notion that deviant IgG-NMDAR subunit expressions in teratomas played a vital role in the pathological process.

### 3.6. The Hypothetical Mechanism of the Pathogenesis and Evidence of Animal Models

Despite the discrepancies on the affected GluN subunits and epitopes, there was a conclusive prevailing theory for mechanism and pathogenesis that all studies had in common. It was generally recognized that the teratoma-associated autoantibodies bound and cross-linked the NMDARs, which altered their surface dynamics and disrupted the interaction with EphB2Rs, along with other synaptic proteins. These antibody-mediated effects eventually caused internalization and degradation of NMDARs (Figure 1C), leading to the reduction in NMDAR density in both synapses and extra-synaptic compartments [27,72,78]. The loss of NMDARs would severely impair synaptic plasticity and NMDAR network function [80]. Therefore, the neural signaling generated by NMDARs was considerably disturbed, potentially causing both neurologic and psychiatric symptoms, which presented as the described anti-NMDAR encephalitis.

The hypothesis seemed to have plausibly explained the pathogenesis of the disease, but this compendious pathophysiology might be too simplistic to reflect the complexity of clinical manifestations in anti-NMDAR encephalitis [54]. In order to scrutinize the interrelation between the mechanisms and symptoms, two well-known animal models were developed [27]. One demonstrated the passive transfer of human NMDAR antibodies, which induced similar changes of reduced synaptic NMDARs and long-term potentiation [81]. The mice eventually presented with memory deficits, anhedonia, depression-like behavior, and a low threshold for seizures which resembled some of the human clinical features, and all these effects were reversible upon discontinuation of the exposure of antibodies [81]. Further, the symptoms could be prevented with an agonist of the EphB2R [82].

In another mouse model of active immunization, the generated antibodies caused a reduction in NMDARs density and NMDAR-mediated currents [83]. Additionally, pathological investigations showed extensive inflammatory infiltrates in company with antibodies, microglial activation, and occasional neuronal loss. The activated microglial cells were consistent with the findings mentioned above in BBB integrity, which neurotoxic metabolites might also account for the injury of neurons. The inspected symptoms of this mouse model were characterized by dramatic hyperactivity, seizures, abnormal motor features (tight circling), and lethargy, also paralleling some symptoms in human disease [83].

Even though synaptic function and plasticity alternations are associated with the above symptoms (Figure 1), the exact comprehensive process causing those terminal manifestations is still not fully understood [81,82,83,84]. Still, be that as it may, the results of these models confirmed that the minuscule changes in synapses and specific immune responses against NMDARs led to a great repository of clinical manifestations in ovarian teratomas with anti-NMDAR encephalitis [27].

## 4. Treatment and Detection for Patients with Ovarian Teratomas and Anti-NMDAR Encephalitis

Since an ovarian teratoma itself was the trigger, tumor removal was deemed a significant part of anti-NMDAR encephalitis management, improving the patients’ outcome and decreasing relapse [48]. In addition, since the immune system also played a crucial role in the pathogenesis, combining both surgery and immunomodulatory treatments was perceived as the most critical management procedure among patients with PNS [27,30]. Conversely, chemotherapy and radiation were not involved in the treatment consideration owing to the resistant nature of MTs [85].

Full recovery and reduced risk of relapse occurred more often in anti-NMDAR encephalitis patients who had an ovarian teratoma and received surgical removal than those without an ovarian teratoma [31]. Regarding the timing of surgery, Dai et al. suggested that early surgical removal of tumors was important for relieving severe neurological conditions [30]. Many neurologists believe that ovarian teratomas should be removed punctually once detected. Systemic and neurologic complications should not be regarded as contraindications for surgery [86]. Nonetheless, 30–60% of anti-NMDAR encephalitis cases in women of childbearing age were associated with the presence of ovarian teratomas, whose removal was crucial in the resolution of symptomatology [87]. Mizutamari et al. reported a successful outcome following detecting and removing a very small ovarian teratoma associated with anti-NMDAR encephalitis during pregnancy [88]. Thus, examinations in search of ovarian teratomas must be envisaged by gynecologists who might have a decisive role in the etiological management and noted that the reproductive function could be preserved through fertility-sparing surgery at the time of removal.

Immunomodulatory methods have been proven to eliminate causes and risk factors, being chief supports and therapeutic options in patients with ovarian teratoma-related encephalitis [89]. The immunotherapy approach and escalation to these patients started with first-line therapies including steroids, intravenous immunoglobulins, or plasma exchange, and transitioned to second-line therapies such as rituximab (chimeric monoclonal antibody against CD20) or cyclophosphamide (alkylating agent) if needed [27,43]. If patients were refractory to these approaches (around 10%), third-line treatments such as bortezomib (a proteasome inhibitor) or tocilizumab (an interleukin-6 receptor antagonist) have been suggested [90]. Interestingly, despite the severity of the disease, most patients would get a response to immunotherapy [91]. Such immunosuppressive management could not cure the disease if used alone [32]. In contrast, removing teratomas might be curative, but only when combining both surgery and immunotherapy could lead to the maximum effect of a brisk clinical reaction and full recuperation [89]. Therefore, the detection and recognition of concurrent ovarian teratoma and anti-NMDAR encephalitis were important.

Regarding the detection of encephalitis, patients usually presented with acute onset of neuropsychiatric symptoms and unresponsiveness to antipsychotic medications [43]. Laboratory studies (serum and CSF) and images, including pelvic ultrasound, magnetic resonance imaging (MRI), computerized tomography (CT), and *electroencephalogram (EEG)*, should be obtained to proceed with the differential diagnosis. Notably, although 67% of patients might have a normal brain MRI, 90% of anti-NMDAR encephalitis patients showed EEG abnormalities [32]. In some cases, integrated ^18^F-*fluorodeoxyglucose positron emission tomography MRI* could be a multimodality approach to evaluate anti-NMDAR encephalitis brain changes and screen with good accuracy possible associated tumors or malignancies [92].

The detection and diagnostic approaches to anti-NMDAR encephalitis-related ovarian teratomas were based on the criteria in Table 3. In particular, patients with anti-NMDAR encephalitis and ovarian teratomas tend to present more severe neurological conditions [30]. Due to the complex and non-specific symptoms, anti-NMDAR encephalitis was commonly mistaken as viral encephalitis, primary psychiatric disorder, drug abuse, neuroleptic malignant syndrome, etc. [91], making it strenuous and challenging to distinguish from others at initial presentation. Currently, the only specific diagnostic test of anti-NMDAR encephalitis is to demonstrate IgG autoantibodies against the GluN1 subunit of the receptors in the patient’s CSF or blood [27,43]. Although the autoantibodies were specific and crucial for a conclusive diagnosis, Thomas et al. reported a case whose anti-NMDAR antibody was negative [93], notifying that clinicians should not defer investigating suspected cases with anti-NMDAR encephalitis and ovarian teratomas even if the antibody was negative.

## 5. Outcome and Associations of Ovarian Teratoma-Related Anti-NMDAR Encephalitis

To evaluate the anti-NMDAR encephalitis outcome, most scientists had reached a consensus on using the mRS (Appendix A), as we discussed in the preceding section [30,32]. While relapse was defined as an mRS score increased ≥1 after at least two months of improvement or stabilization, an outcome was considered good with an mRS score of 0 to 2. Previous studies reported prompt treatment and paucity of intensive care unit (ICU) admission as good outcome predictors [27,30,32]. Herein, we re-emphasize the importance of early tumor excision and immunotherapy administration, which bring out a favorable clinical outcome.

On the contrary, a 5-tiered model was developed to predict the poor outcome of autoimmune encephalitis [94]. The five independent predictors were (1) admission to an ICU, (2) treatment delay of more than four weeks, (3) absence of improvement within four weeks of starting treatment, (4) abnormal MRI, and (5) white blood cell count > 20 cells/μL in CSF. These five variables were assigned 1 point each to construct a total score, called the anti-NMDAR Encephalitis One-Year Functional Status (NEOS) score [27]. The NEOS score strongly correlated with the probability of poor functional status at one year, with 0–1 point for 3% and 4–5 points for 69% [27]. Notably, the score should not be used to guide decisions about withdrawal of care but to help estimate the velocity of clinical improvement and the probably expected outcome [27].

Notably, failure to improve after tumor removal could be a poor prognostic factor for patients with ovarian teratoma-related anti-NMDAR encephalitis [95]. Although more than 75% of patients with anti-NMDAR encephalitis were fully recovered or had only mild sequelae after tumor resection, the remainder experienced severe disability [95]. It remained unknown why certain cases had refractory clinical disease courses, although unsuccessful improvement after tumor resection might be associated with recurrent ovarian teratoma in patients with refractory anti-NMDAR encephalitis [95].

Since ovarian teratomas were the most common female GCTs, we were curious whether there was a risk stratification on female GCTs. Due to their rarity, little was known about prognostic factors and outcomes in female GCTs [85]. Furthermore, management was primarily based on studies of epithelial ovarian cancer and male GCTs. Meisel et al. proposed a modified International Germ Cell Cancer Collaborative Group (IGCCCG) classification as a prognostic assessment on female GCTs [85]. However, the histological characteristics of teratomas involved in that study were mainly immature teratomas without available tumor markers [85], and a sizeable case–cohort study would be required to investigate its validity.

Several studies had shown the race prevalence among anti-NMDAR encephalitis that Asians and Africans were more likely to have a teratoma than White or Hispanic patients [32,39]. Moreover, with a deeper understanding of genetics studies, HLA-A and HLA-DRB1 might play a unique role in ovarian teratoma-associated anti-NMDAR encephalitis [25,96], yet the association of both alleles with disease susceptibility was weak and needed more confirmation [25,27].

Lastly, the disease association between ovarian teratoma and anti-NMDAR encephalitis has not pertained explicitly to both of them. Ovarian mucinous cystadenoma could also cause anti-NMDAR encephalitis, indicating that tumors in anti-NMDAR encephalitis might not be limited to ovarian teratomas [97]. Apart from anti-NMDAR encephalitis, ovarian teratoma could also result in opsoclonus-ataxia syndrome, characterized by involuntary multidirectional ocular saccades and a loss of voluntary coordination [98]. The opsoclonus-ataxia syndrome belongs to PNS as well. Interestingly, behavioral disturbances and involuntary muscle movements might be seen in opsoclonus-ataxia syndrome, similar to anti-NMDAR encephalitis.

## 6. Conclusions

Since first reported 14 years ago, anti-NMDAR encephalitis has gained a great of attention and become recognized as a rare but treatable autoimmune disorder, with about 80% female patients and a striking association with ovarian teratoma. This article reviewed studies from 2007 to 2020 and concluded some information regarding clinicopathological characteristics, such as age distribution, histological property, symptoms at presentation, and relapse rate. Though this entity’s incidence varies in different studies, we have concluded that the overall incidence of ovarian teratoma among anti-NMDAR encephalitis patients is 37.4%. However, little data shows the incidence rate of anti-NMDAR encephalitis in ovarian teratoma patients and, therefore, needs more future study.

We have emphasized the cellular and molecular pathways of the disease foundation and thus proposed a hypothetical mechanism of the pathogenesis in ovarian teratoma-related anti-NMDAR encephalitis. The pathological process involves an eccentric cellular composition of the teratoma, patients’ usual immune response, and the interfered interaction of receptors and proteins, which lead to autoantibody-induced encephalitis. Treatment, detection, and disease outcome are integratively reviewed in this article, while prompt management and cautious survey are needed in suspected patients. Though ovarian teratoma-associated anti-NMDAR encephalitis is uncommon and challenging to diagnose, the growing interests by physician-scientists and the potentially curable nature could be worth investing in research and study to better understand the underlying mechanism.

## Figures and Tables

**Figure 1 ijms-22-10911-f001:**
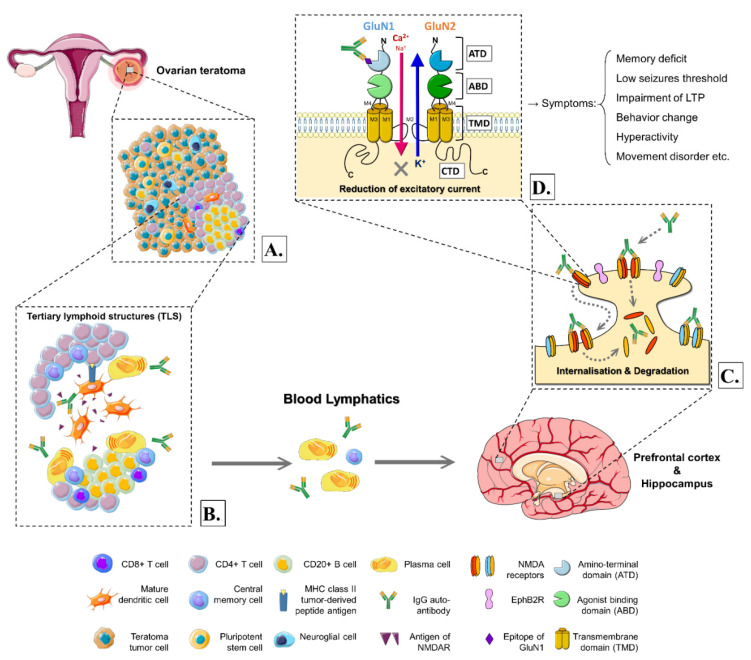
The hypothetical mechanisms of the pathogenesis in ovarian teratoma-related anti-NMDAR encephalitis. (**A**) Ovarian teratomas have a cellular composition of teratoma tumors cells, some sporadic neuroglial cells, and inflammatory infiltrates as well as a tertiary lymphoid structure (TLS) with the germinal center. The NMDARs are expressed on the surface of ovarian teratoma cells. (**B**) The TLS of ovarian teratoma comprises CD4+ T cell zone, CD20+ B cell zone, plasma cells, autoantibodies against NMDARs, central memory cells, and mature dendritic cells. The mature dendritic cells capture neural antigens of NMDARs and present antigenic fragments to CD4^+^ T cells through the MHC class II complex, resulting in the induction of T cells activation, differentiation, and proliferation. Activated CD4^+^ T cells then induced the differentiation of B cells into plasma cells and subsequently generated IgG autoantibodies [20,27]. Eventually, the immunocytes and autoantibodies circulate in the bloodstream and lymphatic systems and cross the blood–brain barrier into CSF. (**C**) The autoantibodies primarily target the hippocampus and prefrontal cortex of the brain, causing antibody-mediated injury to neurons. The autoantibodies bind and induce cross-linking of the NMDARs, altering the surface dynamics of NMDARs and disrupting the interaction with synaptic proteins such as Ephrin-B2 receptor (EphB2R). These antibody-mediated reactions eventually caused internalization and degradation of NMDARs, reducing NMDAR density in both synapses and extra-synaptic compartments. (**D**) The founding components of an NMDAR comprise four domains: an extracellular amino-terminal domain (ATD), a bi-lobed agonist binding domain (ABD), a pore-forming transmembrane domain (TMD), and an intracellular carboxyl-terminal domain (CTD).

**Table 1 ijms-22-10911-t001:** Autoantibodies specific to intracellular antigens associated with paraneoplastic neurological syndromes [20].

Antigen	Autoantibody	Main Neurological Syndromes	Cancer Types
HuD	Hu-IgG (ANNA1)	Sensory neuronopathy, limbic encephalitis, and cerebellar ataxia.	SCLC, NSCLC, and extra-thoracic cancers.
Cdr-2	Yo-IgG (PCA1)	PCD (majority), brainstem encephalitis, and myelopathy.	Ovarian, breast and fallopian tube carcinoma; gastrointestinal cancer in males.
SOX1	SOX1-IgG	LEMS, PCD, and limbic encephalitis	SCLC, NSCLC, and extra-thoracic cancers.
Unknown	ANNA-3	Limbic encephalitis, neuropathies, cerebellar ataxia, myelopathy, and brain stem encephalitis.	SCLC, NSCLC, and other tobacco-related airway cancers.
NOVA1 and NOVA2	Ri-IgG (ANNA2)	Brainstem encephalitis, opsoclonus, laryngospasm, and jaw dystonia	Breast, lung, and neuroblastoma.
Ma1 and Ma2	Ma and/or Ma2-IgG	Limbic encephalitis and brain stem encephalitis	Testicular, lung, and others (mainly gastrointestinal).
ZIC4	ZIC4-IgG	PCD, and others.	SCLC, and ovarian adenocarcinoma.

ANNA: antineuronal nuclear antibody; Cdr-2: cerebellar degeneration-related protein; HuD: Hu-antigen D; LEMS: Lamber-Eaton myasthenic syndrome; Ma: antibodies that react with both Ma1 and Ma2; NOVA: neuro-oncological ventral antigen; NSCLC: non-small-cell lung cancer; PCA: Purkinje cell cytoplasmic antibody; PCD: paraneoplastic cerebellar degeneration; SCLC: small cell lung carcinoma; ZIC4: Zinc finger protein 4.

**Table 2 ijms-22-10911-t002:** Autoantibodies specific to plasma membrane antigens associated with paraneoplastic neurological syndromes [20].

Antigen	Autoantibody	Main Neurological Syndromes	Tumor Types	Frequency of Tumor
DNER	DNER-IgG (PCA-Tr)	PCD.	Hodgkin lymphoma.	>95%
GluA1, GluA2	AMPAR-IgG	Limbic encephalitis.	SCLC, NSCLC, breast, and thymoma.	60–70%
P/Q-type VGCC	P/Q-type VGCC-IgG	LEMS, and PCD.	SCLC.	60%
β1 subunits	GABAbR-IgG	Limbic encephalitis, isolated status epilepticus, cerebellar ataxia, and opsoclonus myoclonus.	SCLC, thymoma, and extra-thoracic cancers.	60%
GluD2	GluD2-IgG	Opsoclonus myoclonus ataxia syndrome.	Neuroblastoma, and ovarian teratoma.	50%
α1, β3, γ2 subunits	GABAaR-IgG	Encephalitis with seizures, cognitive impairment, and behavior changes.	Thymoma, and Hodgkin lymphoma.	40%
GluN1	NMDAR-IgG	Encephalitis with initial psychiatric disturbances, followed by catatonia, dystonia, seizures, aphasia, coma, and central hypoventilation.	Ovarian teratoma.	20–40%
Muscle AChR	Anti-AChR	Myasthenia gravis.	Thymoma.	15%
mGluR1	mGluR1-IgG	Cerebellar ataxia.	Hematologic malignancies, and prostate adenocarcinoma.	10%
DPPX/Kv4.2	DPPX-IgG	Diarrhea, weight loss, cognitive dysfunction, and CNS hyperexcitability.	B-cell neoplasms.	10%
Aquaporin-4	Aquaporin-4-IgG	Neuromyelitis optica spectrum disorders (optic neuritis, longitudinally extensive transverse myelitis, and area postrema syndrome).	Thymoma, breast, and lung.	5%

AChR: acetylcholine receptor; AMPAR: α-amino-3-hydroxy-5methyl-4-isoxazolepropionic acid receptor; DNER: delta and notch-like epidermal growth factor-related receptor; CNS: central nervous system; DPPX: dipeptidyl-peptidase-like protein 6; GABAaR: GABA type A receptor; GABAbR: GABA type B receptor; GluA1: glutamate receptor 1; GluA2: glutamate receptor 2; GluN1: glutamate receptor 1; GluD2: glutamate receptor δ2; LEMS: Lambert–Eaton myasthenic syndrome; mGlur1: metabotropic glutamate receptor 1; NMDAR: N-methyl-D-aspartate receptor; NSCLC: non-small-cell lung cancer; PCD: paraneoplastic cerebellar degeneration; SCLC: small-cell lung carcinoma; VGCC: voltage-gated calcium channel.

**Table 3 ijms-22-10911-t003:** Diagnostic criteria of anti-NMDAR encephalitis [27].

	Criteria	
**Probable Case**	Rapid onset (<3 months) of at least 4 of the 6 major groups of symptoms:	Major groups of symptoms: Abnormal (psychiatric) behavior or cognitive dysfunction; Speech dysfunction (pressured speech, verbal reduction, or mutism); Seizures; Movement disorder, dyskinesias, rigidity, or abnormal postures; Decreased consciousness; Autonomic dysfunction or central hypoventilation.
	Additionally, at least one of the laboratory studies:	Abnormal EEG (focal or diffuse slow or disorganized activity, epileptic activity, or extreme delta brush); CSF with pleocytosis or oligoclonal bands.
	Or 3 of the above groups of symptoms and identification of a systemic teratoma	
	Exclude the recent history of herpes simplex virus encephalitis or Japanese B encephalitis, resulting in relapsing immune-mediated neurological symptoms.	
**Definite Case**	One or more of the 6 major groups of symptoms and **IgG GluN1 antibodies** (antibody testing should include CSF); if only serum is available, confirmatory tests should be included (e.g., live neurons or tissue immunohistochemistry, in addition to a cell-based assay)	
	Exclude the recent history of herpes simplex virus encephalitis or Japanese B encephalitis, resulting in relapsing immune-mediated neurological symptoms.	

**Table 4 ijms-22-10911-t004:** Studies from 2007 to 2020 on the association and characteristics of female patients with ovarian teratomas and anti-NMDAR encephalitis.

	Author/Reference	Study Design	GCT Case Number/Age	Ovarian GCT	Size and Laterality/ Time to Diagnosis	Prodromal Symptoms and PNS	Encephalitis Relapse Rate	GCT Incidence Rate	Note
1	Florance et al. [31]	Single institutional observational study.	32 patients (>18 y/o: 24 cases; ≤18 y/o: 8 cases)	Teratomas, subtype not mentioned.	N/A	Prodromal symptoms: 48%^$^ Autonomic instability: 86%^$^ Movement disorder: 84%^$^Seizures: 77%^$^ Behavior/personality: 59%^$^	0%	32/69 (46.4%)	69 female patients with anti-NMDAR encephalitis (>18 y/o: 43 cases; ≤18 y/o: 26 cases). $: the percentages were based on cases assessable in the original study.
2	Titulaer et al. [32]	Multi-institutional observational study.	207 patients (<12 y/o: 4 cases; 12–44 y/o: 199 cases; ≥45 y/o: 4 cases)	Teratomas, subtype not mentioned.	N/A	Prodromal symptoms: N/A Behavior/cognition: >95% Movement disorder: ~80% Seizures/memory deficits/ speech disorder: 70~80%	12/207 (5.8%)	207/468 (44.2%)	Among 468 female patients with anti-NMDAR encephalitis, 207 had ovarian teratomas, 4 had extraovarian teratomas, and 9 had other tumors.
3	Bost et al. [33]	Single-center retrospective observational study.	51 patients; MT: the median age of 25 years (range: 15–45) IT: the median age of 22 years (range: 12–38)	MTs: 45 cases; ITs: 6 cases (grade 1: 2 cases, grade 2: 3 cases, grade 3: 1 case).	Median (range):MT: 7 days (−26~643); IT: 0 day (−6~131).	Prodromal symptoms: N/A Behavior/personality: 82%^-^ Cognition: 32~100%^-^ Movement disorder: 12~78%^-^ Autonomic instability: 2~74%^-^ Seizures: 8~64%^-^	1/6 (16.7%)^+^	51/195 (26.2%)	There were 195 female patients with anti-NMDAR encephalitis (169 patients were above 12 years of age). +: among the 6 ITs patients −: percentages range from first and subsequent symptoms
4	Dai et al. [30]	A single-center prospective study.	29 patients (mean age: 23.1, range: 10–36)	MTs: 28 cases; ITs: 1 case (grade 1).	Mean size: 4.6 cm (1–12 cm).	Prodromal symptoms: 53%; Mental/behavioral disorder: 89.7%; Seizures: 79.3%; Decreased consciousness: 65.5%; Hyperhidrosis: 62.1%; Speech disorder: 55.2%.	4/29 (13.8%)	29/108 (26.9%)	There were 108 female patients with anti-NMDAR encephalitis and a mean age of 23.4 years (range: 5–72).
5	Zhang et al. [34]	Multi-institutional observational study.	26 patients (mean age: 23.3, range: 14–36)	MTs: 23 cases ITs: 3 cases	N/A	Prodromal symptoms: 38%; Psychiatric symptoms: 92.3%; Speech dysfunction: 84.6%; Seizures: 80.8%; Movement disorder: 76.9%; Decreased consciousness: 61.5%.	3/26 (11.5%)	26/56 (46.4%)	There were 56 female patients with anti-NMDAR encephalitis
6	Yaguchi et al. [35]	Case series.	4 patients (mean age: 28, range: 23–31)	MTs: 3 cases. ITs: 1 case.	N/A	Prodromal symptoms: N/A; Psychosis: 100%; Seizures: 100%; Status epilepticus: 100%. (present 4 out of 4)	0%	4/343 (1.17%)#	All ovarian teratomas: 343 patients; MTs: 327 cases; ITs: 16 cases; (131 of 343 ovarian teratomas had neuroectodermal tissue.) #: incidence of encephalitis among patients with teratomas.
								4/6 (66.7%)	There were 6 female patients with anti-NMDAR encephalitis. Four of them had ovarian teratomas.
7	Xu et al. [36]	Single-center prospective study.	42 patients (>18 y/o: 33 cases; ≤18 y/o: 9 cases)	Teratomas, subtype not mentioned.	N/A	Prodromal symptoms: 48.2%; Psychosis: 82.7%; Seizures: 80.9%; Decreased consciousness: 53.2%; Memory deficit: 48.2%; Speech disturbance: 45.5%.	5/42 (11.9%)	42/143 (29.4%)	There were 143 female patients with anti-NMDAR encephalitis (>18 y/o: 102 cases; ≤18 y/o: 41 cases).
8	Acién et al. [37]	A systematic review of reported cases.	174 patients (mean age: 23.9, range: 7–54)	MTs: 99 cases; ITs: 29 cases; Mixed MTs and ITs: 6 cases; Unknown: 40 cases.	Mean size: 6.7 cm (range: 0.1–22 cm); Right: 56 cases; Left: 46 cases; Bilateral: 20 case; Unknown: 52 cases; Mean time: 28 days (3 to 455 days).	N/A	N/A	N/A	Collected cases and data before 2014.
9	Chiu et al. [38]	Case series.	5 patients (mean age: 18.6, range: 7–28)	MTs: 5 cases; Ovarian fibroma: 1 case.	Mean size: 2.65 cm (2–3.3 cm); Right side: 1 case; Left side: 1 case; Not marked: 3 cases; Mean time: 59.6 days (7 to 150 days).	Prodromal symptoms: 60%; Autonomic dysfunction: 80%; Psychosis: 80%; Seizures: 80%; Decreased consciousness: 60%; Impaired speech: 60%.	1/5 (20%)	5/13 (38.5%)	13 female patients with anti-NMDAR encephalitis (mean age: 19.9, range: 7–28).
10	Yan et al. [39]	Case report and literature review.	15 patients (mean age: 21, range: 7–33)	MTs: 10 cases; ITs: 2 cases; Unknown: 3 cases. (5 of 10 MTs had mature brain tissues.)	Right: 5 cases; Left: 2 cases; Bilateral: 1 case; Unknown: 7 cases.	Prodromal symptoms: 66.7%; Nervous and mental symptoms: 93.3%; Seizures: 53.3%; Dyskinesia: 40%; Autonomic instability: 38.5%.	N/A	N/A	14 published case reports during 2010 to 2019 and one presented case. NMDAR-Ab was positive in CSF.
11	Yu et al. [40]	Case series.	6 patients (mean age: 25, range: 21–27)	MTs: 6 cases (4 of 6 had mature brain tissues).	Mean size: 1.73 cm; Right side: 3 cases; Left side: 3 cases; Mean time: 22.5 days (5 days to 3 months).	Prodromal symptoms: 50%; Psychotic symptoms: 100%; Cognitive decline: 100%; Abnormal movement: 100%; Seizures: 100%; Autonomic dysfunction: 66.7%.	N/A	N/A	NMDAR-Ab was positive in CSF.
12	Dalmau et al. [29]	Case series.	12 patients (mean age: 24, range: 14–44)	MTs: 8 cases; ITs: 4 cases.	Mean size: 6.4 cm (1.5–22 cm); Right side: 4 cases; Left side: 6 cases; Bilateral: 1 case; Mediastium: 1 case; Mean time: 8 weeks (3 weeks to 5 months).	Prodromal symptoms: 83.3%; Seizures: 91.7%; Psychiatric symptoms: 83.3%; Memory deficit: 58.3%; Movement disorder: 50%; Decreased consciousness: 41.7%; Impaired speech: 41.7%; Autonomic dysfunction: 33.3%.	0%	N/A	-
13	Ahmad et al. [41]	Case report.	1 patient, 26 y/o	MT: 1 case.	Size: 2.5 cm; Right side.	Prodromal symptoms: 100% Psychosis: 100% Seizures: 100% Movement disorder: 100% Decreased consciousness: 100%	0%	N/A	NMDAR-Ab was positive in CSF.
14	Omata et al. [42]	Case report.	2 patients (mean age: 12.5, range: 11–14)	MTs: 2 cases.	Mean size: 2.8 cm (1.3–5 cm); Left side: 1 case; Bilateral: 1 case.	Prodromal symptoms: N/A Psychotic symptoms: 100% Movement disorder: 100% Abnormal speech: 50%	0%	N/A	NMDAR-Ab was positive in CSF.
15	Mitra et al. [43]	Case report.	1 patient, 22 y/o	MT: 1 case with neural elements resembling white matter.	Size: 1.4 cm; Right side.	Prodromal symptoms: N/A Altered mental status: 100% Psychosis: 100% Autonomic dysfunction: 100% Movement disorder: 100%	0%	N/A	NMDAR-Ab was positive in CSF.
16	Lwin et al. [44]	Case report.	1 patient, 12 y/o	MT: 1 case.	Size: 5 cm; Left side.	Prodromal symptoms: N/A Psychosis/Abnormal behavior: 100% Seizures: 100%	N/A	N/A	NMDAR-Ab was positive in CSF.
17	Lee et al. [45]	Case report.	1 patient, 24 y/o	MT: 1 case with mature brain tissue.	Size: 1 cm; Right side.	Prodromal symptoms: 100% Abnormal movement: 100% Decreased mental status: 100% Autonomic dysfunction: 100%	N/A	N/A	NMDAR-Ab was positive in CSF.
18	Chernyshkova et al. [46]	Case report.	1 patient, 55 y/o	MT: 1 case.	Left side.	Prodromal symptoms: N/A Altered mental status: 100% Autonomic dysfunction: 100% Seizures: 100% Movement disorder: 100%	N/A	N/A	This is a probable case based on Table 3 since autoantibodies in CSF were not detected.
**Total conclusion (for assessable cases)**			Mean age: 23.97; (age range: 7~55)	MTs: 234 cases; ITs: 46 cases; Unknown: 330 cases.	Mean size: 3.48 cm; Right side: 72 cases; Left side: 61 cases; Bilateral: 23 cases.	Prodromal symptoms: 64.7%; Behavior/personality/ Psychosis: 89.8%; Seizures: 82.4%; Movement disorder: 79.6%; Decreased consciousness: 77.1%; Autonomic dysfunction: 70.5%; Speech disorder/memory deficit: 60.4%.	26/367 (7.1%)	396/1058 (37.4%)	

GCT: germ cell tumor; IT: immature teratoma; MT: mature teratoma; N/A: not assessable or not studied; PNS: paraneoplastic neurologic syndrome; y/o: year-old; NMDAR-Ab: N-methyl-D-aspartate receptor antibody; CSF: cerebral spinal fluid.

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
