# Peer review of "The Association of Ovarian Teratoma and Anti-N-Methyl-D-Aspartate Receptor Encephalitis: An Updated Integrative Review"

_ijms, 2021, doi:10.3390/ijms222010911_

Round 1
Reviewer 1 Report
This review provides a comprehensive literature overview about an interesting association between ovarian teratoma and anti-NMDAR encephalitis. Overall, the article is clearly written and provides the authors’ valuable insights on this topic. However, there are several parts that need the authors’ clarification. Please see below for details.
In page 2, ‘MTs.. are the most common ovarian tumors in women’s second and third decade of life’. This sentence does not sound correct since serous ovarian carcinoma is the most common ovarian cancer. Please correct it accordingly.
Since Table 4 discusses both teratoma/GCT and anti-NMDAR encephalitis, it might be better to include the disease name in each column to prevent the readers’ confusion. For example, include ‘GCT’ in front of the case number/age, relapse, and incidence. In study 1, what does $ indicate and why is the GCT case number different (31 assessable cases vs 32 patients)? In the total conclusion section, why are the GCT numbers different: 280 (234+46) teratomas (column 3) vs 367 (column 6) vs 396 (column 7)?
In Table 4, it is not clear what the relevance of studies #8, 10-18 is since these studies do not have any information about anti-NMDAR encephalitis and thus cannot show any association between ovarian teratoma and anti-NMDAR encephalitis. Although psychiatric symptoms in these teratoma patients are appreciated, it is not clear whether they were diagnosed with PNS or anti-NMDAR encephalitis as there is no information about autoantibodies, one of the criteria to diagnose PNS. Also, they are from the report of one patient’s case.
Are there any studies to determine statistical significance and/or Pearson correlation to demonstrate a positive correlation between ovarian teratoma and anti-NMDAR encephalitis?
How were p values calculated in Table S2? Did the authors calculate these p values or did the original authors of ref 56 calculate these p values?
Are there any common genetic events (e.g. SNP or mutation in specific genes) between ovarian teratoma and anti-NMDAR encephalitis? Are there any genetic factors or signaling pathways that might be responsible for both ovarian teratoma and anti-NMDAR encephalitis? If so, it would be helpful to include this information in one of the mechanism sections.
In page 17, please indicate what molecules rituximab and cyclophosphamide inhibit as the authors did for bortezomib (a protease inhibitor).
The authors proposed a hypothetical mechanism of the pathogenesis in ovarian teratoma-related anti-NMDAR encephalitis. However, the authors noted that ovarian mucinous cystadenoma could also cause anti-NMDAR encephalitis (p19). Will this hypothetical mechanism also explain this mucinous cancer-related anti-NMDAR encephalitis? Or are there multiple mechanisms other than the authors’ proposed mechanism that could cause anti-NMDAR encephalitis?
Author Response
Dear Editor in Chief and our respectful reviewers
We are really grateful for your expertise and sincere comments on our manuscript. All your comments are professional and helpful to the review article. The followings are our replies and all the revised parts would be highlighted in red in our revised manuscript.
Reviewer 1:
This review provides a comprehensive literature overview about an interesting association between ovarian teratoma and anti-NMDAR encephalitis. Overall, the article is clearly written and provides the authors’ valuable insights on this topic. However, there are several parts that need the authors’ clarification. Please see below for details.
- In page 2, ‘MTs.. are the most common ovarian tumors in women’s second and third decade of life’. This sentence does not sound correct since serous ovarian carcinoma is the most common ovarian cancer. Please correct it accordingly.
Thank you for your kind words and professional comments. We missed typing ‘germ cell’ ahead of ‘tumors’. The sentence should be ‘MTs.. are the most common ovarian germ cell tumors in women’s second and third decade of life’. We have revised the sentence and highlighted them in red.
- Since Table 4 discusses both teratoma/GCT and anti-NMDAR encephalitis, it might be better to include the disease name in each column to prevent the readers’ confusion. For example, include ‘GCT’ in front of the case number/age, relapse, and incidence. In study 1, what does $ indicate and why is the GCT case number different (31 assessable cases vs 32 patients)? In the total conclusion section, why are the GCT numbers different: 280 (234+46) teratomas (column 3) vs 367 (column 6) vs 396 (column 7)?
Thank you for your professional comments.
1) We sincerely agreed with the recommendation and put the disease names (GCT or encephalitis) in each column to clarify the meaning. Thus we revised “GCT case number/age” in column 3, “Encephalitis relapse rate” (which means the encephalitis relapsed among patients with anti-NMDAR encephalitis and ovarian teratoma) in column 7, and “GCT incidence rate” (which means the incidence of ovarian teratoma among patients with anti-NMDAR encephalitis) in column 8.
2) For study 1, $ indicates the percentages based on 31 cases with assessable medical records for prodromal symptoms in the original study, although the study had 32 female patients with anti-NMDAR encephalitis and GCT. In the original study of Florance et al. [31], its table of clinical features showed “prodromal symptoms from 31 assessable cases”. The article also mentioned that some data were assessable in 31 patients, and one was lost to follow-up. Therefore, we have changed “31 assessable cases” to “cases assessable in the original study” to avoid confusion.
3) In the total conclusion section, it is important to elucidate that column 3, column 6, and column 7 are calculated independently with different interpretations in each column. Column 3 only shows the total number of different type teratomas (MT, IT, and unknown type) we collected from all the studies (total number of teratomas: 234+46+330=660). For column 6, it indicates some studies that had data of encephalitis relapse (not all the 18 studies we listed) and concludes a generalized rate from cases assessable. As for column 7, it comes from the collection of studies that included incidence rates and is a total incidence rate from cases assessable.
To better clarify, for instance, study #7 had 42 patients with teratomas of unknown type; hence those 42 patients would not be included in the 280 (234+46) cases of teratomas (yet be included in 330 cases of unknown). However, study #7 had provided data with encephalitis relapse and CGT incidence rate, so those 42 patients would be calculated in the total encephalitis relapse and total GCT incidence rate.
At last, we want to reiterate that column 3, column 6, and column 7 are calculated separately based on the assessable studies.
- In Table 4, it is not clear what the relevance of studies #8, 10-18 is since these studies do not have any information about anti-NMDAR encephalitis and thus cannot show any association between ovarian teratoma and anti-NMDAR encephalitis. Although psychiatric symptoms in these teratoma patients are appreciated, it is not clear whether they were diagnosed with PNS or anti-NMDAR encephalitis as there is no information about autoantibodies, one of the criteria to diagnose PNS. Also, they are from the report of one patient’s case.
Thank you for your professional comments. We have thoroughly reviewed studies #8, 10-18 again, and all but one study (#18) had noted that anti-NMDAR antibodies were positive in the cerebrospinal fluid (CSF). Though these studies did not state the presence of “IgG GluN1 antibodies”, we consider the established diagnosis of anti-NMDAR encephalitis appreciable. Moreover, the diagnostic criteria we listed in Table 3 were developed in 2016. Thus studies before 2016 might not have a consentaneous guideline to follow, making the diagnostic approach had its difficulties. To clarify this issue, we added the information on anti-NMDAR antibodies in the note column of each study in Table 4.
The only one patient in study #18 was indeed a “probable case” based on the criteria of Table 3 since antibodies in CSF were not detected. However, the original authors had proposed the concept of “seronegative autoimmune encephalitis,” describing cases that symptoms indicated autoimmune encephalitis, yet the CSF and blood analysis was negative for antibodies. The authors had explained several causes, one of which was due to technical limitations and a subclinical picture of initial symptoms. The proper diagnosis was delayed, and by the time diagnosis was suspected, the number of antibodies had decreased below the detection threshold. Especially in older individuals (the patient’s age: 55), the penetrability of the brain-blood barrier increases, which allows even undetectable levels of antibodies to cross and impede brain function. In study #18, the older age of onset and the initial seronegativity of the CSF led to an ambiguous diagnosis. Nevertheless, several factors led them to conclude the diagnosis of probable NMDAR encephalitis, including ovarian teratoma, therapeutic response to immunotherapy, and significant improvement of symptoms after tumor removal.
- Are there any studies to determine statistical significance and/or Pearson correlation to demonstrate a positive correlation between ovarian teratoma and anti-NMDAR encephalitis?
Thank you for your professional comments. Since all articles we reviewed are case reports and case series, it is a pity that they did not perform any within-group and between-group correlation analyses between ovarian teratoma and anti-NMDAR encephalitis. A large-scale cohort study might be needed to elucidate whether there is a positive correlation.
- How were p values calculated in Table S2? Did the authors calculate these p values or did the original authors of ref 56 calculate these p values?
Thank you for your professional comments. Table S2 was modified from table 1 of our reference 56 (Am J Surg Pathol 2019, 43, 949-964). All data were selectively adopted from the original study, and the original authors calculated the p values.
- Are there any common genetic events (e.g. SNP or mutation in specific genes) between ovarian teratoma and anti-NMDAR encephalitis? Are there any genetic factors or signaling pathways that might be responsible for both ovarian teratoma and anti-NMDAR encephalitis? If so, it would be helpful to include this information in one of the mechanism sections.
Thank you for your professional comments. We have thoroughly reviewed the latest update of genetic events between ovarian teratoma and anti-NMDAR encephalitis again, and the only article related was our reference [25] “Genetic predisposition in anti-LGI1 and anti-NMDA receptor encephalitis”. Mueller et al. demonstrated a genome-wide-association study in 1,194 controls and 150 patients with anti-N-methyl-D-aspartate receptor (anti-NMDAR, n=96) or anti-leucine-rich glioma-inactivated1 (anti-LGI1, n=54) autoimmune encephalitis. Among those encephalitis cases, anti-LGI1 encephalitis was highly associated with 27 SNPs
in the HLA-II region. On the other hand, the anti-NMDAR encephalitis, our major focus in this review, yielded no genome-wide significant associations in SNP-wise analysis. Furthermore, patients of anti-NMDAR encephalitis have only a minor association with HLA-I allele B*07:02 (P=0.039), as we discussed on pages 18 and 19. The association of alleles or genetic events with disease susceptibility was weak and needed more confirmation or be validated in larger cohorts.
Therefore, according to the latest study assessable, we have concluded that there are no explicit genetic factors or signaling pathways that might be responsible for both ovarian teratoma and anti-NMDAR encephalitis.
- In page 17, please indicate what molecules rituximab and cyclophosphamide inhibit as the authors did for bortezomib (a protease inhibitor).
Thank you for your professional comments. According to your kind suggestion, we added that rituximab is a chimeric monoclonal antibody targeted against the pan-B-cell marker CD20, and cyclophosphamide is alkylating agent chemotherapy. The main effect of cyclophosphamide is due to its metabolite phosphoramide mustard. Phosphoramide mustard forms DNA crosslinks both between and within DNA strands. This is irreversible and leads to cell apoptosis.
- The authors proposed a hypothetical mechanism of the pathogenesis in ovarian teratoma-related anti-NMDAR encephalitis. However, the authors noted that ovarian mucinous cystadenoma could also cause anti-NMDAR encephalitis (p19). Will this hypothetical mechanism also explain this mucinous cancer-related anti-NMDAR encephalitis? Or are there multiple mechanisms other than the authors’ proposed mechanism that could cause anti-NMDAR encephalitis?
Thank you for your professional comments. In Table 2 we have listed various tumor types which may be associated with autoantibody-related paraneoplastic neurological syndromes, and anti-NMDAR encephalitis is observed to be associated with ovarian teratomas. The hypothetical mechanism to explain the association is primarily based on the teratomatous elements (especially the neuroglial tissue) and its tumor-associated immunocytes in ovarian teratomas, which may induce T cells activation, differentiation of B cells into plasma cells, and generation of IgG autoantibodies. However, ovarian mucinous cystadenoma does not have teratomatous elements or neuroglial cells. Its mechanism is definitely different from teratoma. Moreover, there is only one case report with ovarian mucinous cystadenoma and anti-NMDAR encephalitis. Its mechanism is challenging to be investigated due to its rarity.

Reviewer 2 Report
This is a well written systematic review covering the epidemiological, molecular and clinical aspects of this important topic.
Author Response
Thank you for your kind words and sincere comments. We appreciate your professional peer review and supports.

Round 2
Reviewer 1 Report
The authors have addressed all concerns and suggestions made by the reviewer and improved the clarity of the manuscript. The revised manuscript provides a comprehensive overview and mechanistic insights underlying the association between ovarian teratoma and anti-NMDAR encephalitis. Thus, it is recommended for the publication.